# Genome-Wide Identification and In Silico Analysis of ZF-HD Transcription Factor Genes in *Zea mays* L.

**DOI:** 10.3390/genes13112112

**Published:** 2022-11-14

**Authors:** Md. Abir Ul Islam, Juthy Abedin Nupur, Muhammad Hayder Bin Khalid, Atta Mohi Ud Din, Muhammad Shafiq, Rana M. Alshegaihi, Qurban Ali, Qurban Ali, Zuha Kamran, Mujahid Manzoor, Muhammad Saleem Haider, Muhammad Adnan Shahid, Hakim Manghwar

**Affiliations:** 1Lushan Botanical Garden, Chinese Academy of Sciences, Jiujiang 332000, China; 2Faculty of Applied Biological Sciences, Gifu University, Gifu 501-1193, Japan; 3Department of Agricultural, Food and Nutritional Science, University of Alberta, Edmonton, AB T6G 2R3, Canada; 4National Research Center of Intercropping, The Islamia University of Bahawalpur, Bahawalpur 63100, Pakistan; 5Maize Research Institute, Sichuan Agricultural University, Chengdu 611130, China; 6Key Laboratory of Crop Physiology Ecology and Production Management, College of Agriculture, Nanjing Agricultural University, Nanjing 210095, China; 7Department of Horticulture, University of the Punjab, Lahore 54000, Pakistan; 8Department of Biology, College of Science, University of Jeddah, Jeddah 21493, Saudi Arabia; 9Department of Plant Breeding and Genetics, University of the Punjab, Lahore 54000, Pakistan; 10Laboratory of Integrated Management of Crop Diseases and Pests, Ministry of Education, Department of Plant Pathology, College of Plant Protection, Nanjing Agricultural University, Nanjing 210095, China; 11Department of Entomology, University of the Punjab, Lahore 54000, Pakistan; 12Department of Plant Pathology, University of the Punjab, Lahore 54000, Pakistan; 13Horticultural Sciences Department, University of Florida/IFAS, North Florida Research and Education Center, Quincy, FL 32351, USA

**Keywords:** maize, *ZHD* genes, abiotic stress, structure, expression profile

## Abstract

Zinc finger-homeodomain proteins are amongst the most prominent transcription factors (TFs) involved in biological processes, such as growth, development, and morphogenesis, and assist plants in alleviating the adverse effects of abiotic and biotic stresses. In the present study, genome-wide identification and expression analyses of the maize *ZHD* gene family were conducted. A total of 21 *ZHD* genes with different physicochemical properties were found distributed on nine chromosomes in maize. Through sequence alignment and phylogenetic analysis, we divided *ZHD* proteins into eight groups that have variations in gene structure, motif distribution, and a conserved ZF domain. Synteny analysis indicated duplication in four pairs of genes and the presence of orthologues of maize in monocots. Ka/Ks ratios suggested that strong pure selection occurred during evolution. Expression profiling revealed that the genes are evenly expressed in different tissues. Most of the genes were found to make a contribution to abiotic stress response, plant growth, and development. Overall, the evolutionary research on exons and introns, motif distributions, and cis-acting regions suggests that these genes play distinct roles in biological processes which may provide a basis for further study of these genes’ functions in other crops.

## 1. Introduction

The zinc finger-homeodomain (ZF-HD) TFs, containing a conserved zinc finger (ZF) domain in the N-terminal and a homeodomain (HD) in the C-terminal, are members of a plant-specific TF superfamily [1]. The ZF-HD gene family plays an important role in plant developmental processes and stress responses. *ZHD* genes are plant-specific, nearly all intronless, and are related to MINI ZINC FINGER genes that possess only the zinc finger. Phylogenetic analysis suggested that ZHDs have expanded considerably during angiosperm evolution [1]. Biotic and abiotic stresses, such as drought, salinity, heavy metals, high temperatures, heat stress, chilling stress, insects, pathogens, mechanical injury, etc., cause severe damage in terms of kmo, plant growth, development, yield, and quality [2,3,4,5,6,7,8,9,10]. A well-specialized gene network encodes numerous proteins that systematically control plant growth. To control plant differential growth, flowering, development, alleviation of the adverse impacts of both abiotic and biotic stresses, signal transduction, and morphogenesis, a special class of proteins called transcription factors (TFs) bind the particular DNA or nucleotide sequences responsible for these functions [11]. TFs help plants endure adverse conditions by regulating the binding of specific promoter cis-elements involved in signalling [12]. TFs are found in a wide range of regulatory proteins that can bind to DNA/RNA sequences and participate actively in protein–protein interactions [2].

A conserved homeodomain, which consists of 60 amino acids and has a characteristic three-helix shape, can interact with a variety of DNA sequences [13]. HD domain proteins are divided into subgroups based on their size, structure, location, and connection with other proteins: WUSCHEL-related HB (WOX), knotted-related HB (KNOX), Bell-type HD, and zinc finger motif-associated HD (ZF-HD), leucine zipper-associated HD (HD-Zip), and HD associated with a finger domain (PHD finger) [14,15]. *ZF-HD* is engaged in signal transduction in plants under various abiotic and biotic conditions and has aroused the curiosity of researchers interested in learning more about this TF’s role in plants. ZF-HD proteins were first identified as potential regulators of the C4 phosphoenolpyruvate carboxylase gene (PEP-Case) in C4 *Flaveriatrinervia* species [16]. An N’-end conserved zinc finger domain with zinc ions, rich in cystine or histidine residues, and a cysteine-rich N′-end conserved zinc-finger domain (ZF) with zinc ions and cysteine or histidine residues are two structural properties of these proteins [17], along with a C′ -end conserved homeodomain (HD) [1]. ZF motifs are surrounded by cysteine or histidine residues, either singly or in pairs, and are stabilized by a zinc ion in the form of a finger-shaped loop [1]. The N-terminal part or ZF domain has two types of domains, CH_2_C and C_3_H_2_, separated by a variable-length spacer. The primary function of the HD domain of the ZF-HD transcription factor is to bind DNA sequences to activate or repress the targeted genes [18]. Protein–DNA interactions mediated by HD are enhanced by ZF domains.

There is clear evidence underscoring that *ZF-HD/ZHD* proteins play a pivotal role in alleviating the adverse effects of environmental stresses [1,19,20,21]. For instance, *Arabidopsis* ZHD4 protein expression is upregulated in case of drought, salinity, and cold stress [22]. Additionally, *AtZHD1* and *AtZHD10* improve drought tolerance [23] and simultaneously modulate hormone signalling [24]. In another model crop, rice, among 14 ZHDs proteins, four respond to cold and drought stress and can bind with the promoter region of the *DREB1* gene family [25]. Other crops, such as Chinese cabbage, have 31 *ZF-HD/ZHD* genes [1]. Recent studies have reported the upregulation of *ZHD* genes under various abiotic stress conditions. For instance, four *ZHDs* in cucumber [26], ten *ZHDs* in wheat [27], *NtZHD21* in tobacco [19], *LlZHD4* in *Lilium lancifolium* [20], and barley’s *HvZHD1* were upregulated under abiotic stresses, such as cold, drought, salt, water deficiency, etc.

Furthermore, *ZF-HD/ZHD* genes can modulate biological processes in plants [14]. For instance, in *Arabidopsis*, *AtZHD5* is responsible for leaf size enlargement, *AtZHD10* is involved in hypocotyl elongation, and *AtZHD8* [24], tomato *SlZF-HD7*, and tartary buckwheat *FtZF-HD11* (*Fagopyrum tataricum*) [28,29] play vital parts in the flowering of these plants. Furthermore, earlier research has revealed that the majority of Chinese cabbage *BraZF-HD* genes and wheat *TaZF-HD*s are involved in biological activities [1,29].

*Z. mays* L. is the most extensively grown cereal crop in Africa and South America. Nowadays, it is becoming popular in developing countries, such as Bangladesh, and in developed countries [30] for use in human and animal consumable products, such as corn syrup, corn starch, baby corn, and feed. The world’s maize production climbed from 313 million tons in 1971 to 1162 million tons in 2020, with an average yearly growth rate of 3.07 percent (https://knoema.com/atlas/World/topics/Agriculture/Crops-Production-Quantity-tonnes/Maize-production) (accessed on 1 September 2020). Abiotic stresses, such as intense water logging, extreme temperature, and drought, affect maize production significantly [31]. Drought, high salinity conditions, and extreme temperatures all cause transcription factors (TFs), such as *ZHD*TF, to interact with cis-elements or other TF proteins to respond to various stresses in signal transduction pathways of stress response which control plant growth and development by protecting plants [32,33]. Since maize crop yield is highly damaged by abiotic stresses, it is essential to identify the *ZF-HD* gene family roles in this crop.

To date, no study has been carried out on the *ZF-HD/ZHD* gene family in maize (*Z. mays*). So far, functional analysis of *ZmZHD9* has been performed to identify the role of the gene in the case of drought stress [34]. We proceeded to adopt a bioinformatics approach for a genome-wide characterization and evolutionary analysis of *ZHD* genes and their encoded proteins in the maize genome [35]. We intended to analyze chromosomal locations, gene structures, promoter elements, evolutionary relationships, distinct tissue expressions, duplication patterns, and miRNA patterns. Our results lay the groundwork for exploring the mechanisms of *ZHD* genes in response to various abiotic stresses in maize.

## 2. Results

### 2.1. Identification of ZHD Family Genes and Sequence Analysis of Their Proteins

In *Z. mays*, twenty-one *ZF-HD* genes were found. The names of the genes identified were chosen from those given by GrassTFDB [36] (Table 1). Out of 10 chromosomes, chromosome number 9 of maize did not possess any one of the 21 ZF-HD domains. The lengths of the protein sequences varied from 89 aa to 655 aa, whereas their molecular weights ranged from 9.8 kDa to 71.6 kDa (Table 1). In both cases, *ZmZHD13* and *ZmZHD18* showed the minimum and maximum amino acid lengths and molecular weights (Table 1). The genes *ZmZHD10*, *ZmZHD12*, *ZmZHD15*, *ZmZHD16,* and *ZmZHD17* are associated with theoretical pI values less than 7, while the rest of them showed values higher than 7 (Table 1), which shows the values where the amino acids can be neutral. The GRAVY values of *ZmZHD*s ranged from −0.985 to −0.138, demonstrating that the proteins are hydrophilic in nature (Table 1). Although all the *ZmZHD* proteins are in the nucleus, some of them are also located in the cell wall and chloroplasts (Table 1). From all the parameters, we can predict that *ZmZHD* proteins have diverse physicochemical properties.

### 2.2. Sequence Alignment and Phylogenetic Tree Construction

To conclude the evolutionary analysis, we performed a multiple sequence alignment and built a phylogenetic tree for maize *ZmZHD* proteins and *ZHD* proteins of other species using protein sequences of 21 maize *ZHD*s, 14 rice *ZHD*s, 15 *Arabidopsis ZHD*s, 16 foxtail millet *ZHD*s, 14 sorghum *ZHD*s, 11 barley *ZHD*s, 21 purple false brome (*Brachypodiumdistachyon*) *ZHD*s, and 13 Heller’s rosette grass (*Dichantheliumoligosanthes*) *ZHD*s. In two multiple sequence alignments (MSAs), MSA1 contains only *ZmZHD* proteins (Figure 1). Motif 1 and Motif 4 are presented, which represent the ZF (zinc finger) domains (Appendix A). *ZHD* proteins are classified into eight groups, I to VIII, according to the popularized tobacco and wheat *ZHD* family classifications [19,37]. Clade 1 is the smallest class, containing about 7.2% of the total 125 ZHD proteins, and Clade VIII is the largest class, containing 22.4% of the total 125 ZHD proteins. (Figure 2). In the largest group, around 21% of maize and barley proteins, about 14% of rice and foxtail millet proteins are present (Figure 2). Clade VII is next to the largest one and contains about 17% of the *ZHD* proteins found in all the mentioned species. Among these, approximately 19% are found in maize, 15% in rice, millet, and sorghum, and 10% in *Arabidopsis* and barley (Figure 2). In Clades III and VIII, all plant proteins are present asymmetrically.

### 2.3. Chromosomal Location, Gene Structure, and Motif Composition Analysis

Twenty-one ZmZHDs were mapped onto 10 *Z. mays* chromosomes, according to their locations (Appendix A). *ZmZHD* genes are not symmetrically distributed across all of the 10 chromosomes. None of the 21 genes is located on chromosome number 9 (Chr9). Chr4 contains four genes, while Chr1 and Chr2 contain 3 genes, and Chr3, Chr5, and Chr10 contain 2 genes each (Appendix A). By inputting the entire lengths of the *ZmZHD* protein sequences, a phylogenetic tree was created to examine the evolutionary connections among the 21 *ZmZHD* genes. This tree was split into two classes, Class I and Class II, which were further divided into four and three subclasses, respectively (Figure 3A). We identified at least two motifs and a maximum of seven motifs in *ZHD* proteins using MEME (Figure 3C). All contain Motif 1 and Motif 4 in their sequences, and these constituted the most highly conserved parts of the ZF domain. Class I genes contain a greater number of motifs than Class II genes. Interestingly, Motif 2 and Motif 3 are present in all the members of class I, while they are only present in Subclass IIb. Motif 9 was detected in the specific subfamily subclass Ic. Almost all the *ZmZHD* genes have no introns in their sequences (Figure 3B). Only about 24% of the genes have introns ranging from 1 to 4, while only two of the genes have UTR regions in their sequences (Figure 3B).

### 2.4. Analysis of Cis-Elements in ZmZHD Promoters

Even though cis-acting elements are non-coding DNA sequences, they influence transcriptomic processes in gene promoter regions. Plant CARE software was used to identify around 25 cis-acting regions from the 2000 bp upstream regions of the genomic sequences of the *ZmZHD* genes (Figure 4). *ZmZHD* genes contain plant-growth- and development-related promoters, such as circadian, the O2- site, as-1, the AAGAA- motif, the CCAAT- motif, the GCN4- motif, and the RY- element (Figure 4). Abscisic acid-responsive ABRE, the MeJA-responsive CGTCA- motif, the gibberellic acid-responsive GARE motif, salicylic acid-responsive TCA-element, TGA-element, and auxin-responsive AUxRR-core are hormone-responsive cis-elements found in the *ZmZHD* promoters (Figure 4). Stress-related components were found in several promoters, including TC-rich repeats implicated in defense and stress response, as well as low-temperature-response (LTR)-related motifs, and MYB and MYB binding sites were found in all *ZmZHD* promoters, these being implicated in the induction of drought, high-salt, and low-temperature responses (Figure 4). These results indicated that *ZHD* genes in maize are mainly responsible for hormonal and biotic and abiotic stress tolerance.

### 2.5. Synteny and Evolutionary Analysis of ZmZHDGenes

Gene duplication is a common occurrence in all organisms that results in the creation of new functional genes from previously existing ones, which drives evolution. As a result, we used Advanced Circos in TBtools to perform a microsynteny analysis to evaluate duplications among the *ZmZHD* genes. In four gene pairs, segmental duplications were discovered (Figure 5). To further investigate the gene duplications in the *ZHD* gene family, a duel synteny analysis was performed including maize and four other plants: sorghum, foxtail millet, the *Oryza indica* group, and *Arabidopsis* (Figure 6). The results showed that all the monocots, i.e., sorghum, foxtail millet, and the *Oryza indica* group, have 21 syntenic relations with maize while *Arabidopsis* has only one (Figure 6). As a result, there was more genetic overlap found between maize and monocot genomes than between *Z. mays* and dicot genomes. In addition, all monocot genes had orthologues in maize, implying that maize has undergone additional whole-genome duplication (WGD) events during its evolution. We computed Ka, Ks, and Ka/Ks values for four homologous *ZmZHD* gene pairs to examine evolutionary limitations and selection pressures on the *ZmZHD* genes (Appendix A). Ks values can be used to retrodict the time of whole-genome duplication (WGD) occurrences, since they indicate the background base substitution rate [38,39]. The Ks values for the *ZmZHD* gene pairs varied from 0.04 to 91.87, implying that a large-scale *ZmZHD* gene duplication event occurred between 7066.56 and 2.88 million years ago (MYA) (Appendix A). The gene pairs’ Ka/Ks ratios were all less than 1.0, suggesting that these genes were subjected to strong purifying selection during evolution.

### 2.6. Construction of a PPI Network and Expression Profiling of ZmZHD Genes in Various Tissues

To predict potential interactions among the proteins, we used the STRING database (https://string-db.org/) accessed on 1 September 2022. Only 8 proteins, *ZmZHD2*, *ZmZHD3*, *ZmZHD3*, *ZmZHD3*, *ZmZHD3*, *ZmZHD3*, *ZmZHD3*, *ZmZHD3*, *ZmZHD5*, *ZmZHD6*, *ZmZHD7*, *ZmZHD11*, *ZmZHD16*, and *ZmZHD20*, out of the 21 were correlated at the medium level (0.400) and at the highest level (0.900) of confidence (Figure 7). Each of the eight proteins is interconnected with the other four. ZmZHD6, for example, is closely linked to *ZmZHD2*, *ZmZHZD5*, *ZmZHD7*, and *ZmZHD20*. The core nodes of *ZmZHD6* and *ZmZHD11* only have high confidence levels for their associated proteins (highest confidence level, 0.900) (Figure 7 and Appendix A).

Specific gene expression patterns in certain developmental activities in plants can usually be predicted with tissue-specific transcriptome data. Among *ZmZHD* genes, *ZmZHD11* is expressed in almost all of the 16 tissues in maize plants. The expression levels of the Group A proteins, *ZmZHD2*, *ZmZHD4*, *ZmZHD11*, and *ZmZHD21*, for the sixteen tissues were high in comparison to those of the others (Figure 8 and Appendix A). The protein expression levels were even for all the tissues, for example, maize unpollinated silk (US), vegetative meristem (VM), pericarp and aleurone (PA), embryo after pollination (EmAP), endosperm after pollination (EnAP), internode (IN), mature leaf (ML), mature female spikelet (MFS), primary root (PR), secondary root (SR), root differentiation zone (RDZ), root elongation zone (REZ), stomatal divisional zone of the leaf (SDZL), tip of ear primordium (TEP), germinated embryo (GEM), and growth zone of leaf (GZL) tissues. We can predict that the expression profiles of the Group A genes are much higher than those of the others.

### 2.7. MiRNA Target Site Prediction and Validation

miRNAs cleave mRNA or inhibit translation to produce proteins and regulate target gene expressions. About 29 miRNA families were found in the maize genome, with 188 members, and 26 of 29 families exhibited perfect and sometimes nearly perfect target sequences. All the mature miRNA sequences were predicted against the CDSs of *ZmZHD* genes and 77 miRNAs were shown to be present (Appendix A). miRNA167 has ten target sites in *ZmZHD5*, miRNA4109 has seven target sites in *ZmZHD1*, and miRN4099 and miRN4186 have four target sites in *ZmZHD10* and *ZmZHD21*, respectively (Figure 9). One miRNA, miR2275, has target sites in different genes, such as *ZmZHD1*, *ZmZHD15*, *ZmZHD1*, *ZmZHD4*, *ZmZHD15*, *ZmZHD20,* and *ZmZHD1*, and most of the rest of the miRNAs have one or two target sites (Appendix A). These results reveal the correlations of Zma-miRNA167 and miRNA4109 with other miRNA families.

## 3. Discussion

A conserved zinc finger (ZF) domain on the N-terminal and a homeodomain on the C-terminal is present in the zinc finger-homeodomain (*ZF-HD*). The homeodomain possesses a highly conserved structure containing approximately 60 amino acids. The homeodomain is folded into a recognition helix, which has a characteristic three-helix structure that is attached to the main sulcus of DNA, forming a special link with DNA [40]. *ZF-HD* transcription factors are only present in plants and play an important role in plant growth and development, as well as biotic and abiotic stress alleviation [41]. About 21 *ZmZHD* genes were found in maize in TFDB and BLASTp searches.

*ZHD* genes are exhibited only in terrestrial plants and expanded during angiosperm evolution [1,42]. Various evolutionary and structural analyses have been performed for the structural analysis of ZmZHD domains. These involved various methods, such as sequence alignment, phylogenetic tree, gene structure, motif organization, synteny, and gene duplication analyses. The function of a gene family is determined by the extent and the types of conserved regions basically present in an appropriate sequence alignment. Multiple sequence alignments showed that Motif 1 and Motif 4, which is popularly known as the ZF dimer (Figure 1), are consistent with those of other plant species [1,42], indicating that ZHD proteins have similar structures. *ZmZHD* genes were categorized into eight groups (I-VIII) in our tree analysis (Figure 2), which is consistent with earlier phylogenetic research on the crops [37,43,44,45,46,47]. Except for Groups I, IV, and VII, *ZF-HD/ZHD* proteins from *Arabidopsis*, rice, and maize were found to be in the majority of the groupings (Figure 2). Maize and rice were found in Groups I and IV, but only *Arabidopsis* and maize were found in Group VII (Figure 2), indicating a protein divergence from both monocots and dicots. Evolutionary insights can be extracted from the gene structures of the gene families [26]. In our gene structure study, most of the genes in the *ZF-HD/ZHD* gene family do not contain introns or UTRs (Figure 3B), and this phenomenon can be observed in many species [21,42,48]; the loss of introns might lead to an immediate response to abiotic stress.

On the contrary, five genes have one to four introns (Figure 3B), indicating the structural divergence of the maize *ZF-HD/ZHD* gene family. Except for tomatoes, our findings imply that *ZF-HD/ZHD* family genes have been largely conserved in evolution, along with their functions. In previous studies of *ZF-HD/ZHD* genes in other plants, researchers have concluded that these genes have undergone severe purifying selection, and their functions cannot be differentiated [42,48,49,50,51,52]. Motif analysis showed various conserved motifs of *ZmZHD* proteins in Classes I and II, revealing similar motifs present in the same subclass in the phylogenetic tree that are functionally similar (Figure 3A,C). Gene duplication mechanisms, such as segmental duplication, tandem repeats, and retro- and/or replicate transposition, contributed to biological evolution [53]. Many gene families have been documented to have expanded as a result of segmental duplication [54]. The gene pairs’ Ka/Ks ratios indicate that they have undergone purifying selection during genome-wide evolution, and the *ZmZHD* gene family’s duplication times vary from 2.88 to 7066.56 MYA (Appendix A).

Abiotic stresses adversely affect the growth and development of maize and ultimately affect economic production. ZF-HD/ZHD TFs play pivotal roles in the biological processes of plants [28]. For instance, in *Arabidopsis*, overexpressed ZF-HD1 upregulated several stress-inducible genes, eventually leading to significant increases in drought resistance [23]. So far, in maize crops, the expression patterns of *ZmZHD*s under abiotic stress have not yet been properly investigated. TF mechanisms depend on the cis-elements present in the related genes, which actually regulate the stress signalling and expression of the responsible genes. Numerous cis-elements were identified in a promoter region analysis of plant hormones and abiotic stresses (Figure 4), reflecting the *ZmZHD* gene expression roles in the external environment. Abscisic acid-responsive ABRE, MeJA-responsive CGTCA-motif, gibberellic acid responsive GARE-motif, salicylic acid-responsive TCA-element, TGA-element, and auxin-responsive AUxRR-core are hormone responsive cis-elements found in the *ZmZHD* promoters (Figure 4). Stress-related components were found in several promoters, including TC-rich repeats implicated in defense and stress response, as well as low-temperature response (LTR)-related motifs, and MYB and MYB binding sites were found in all *ZmZHD* promoters, these being implicated in the induction of drought, high-salt, and low-temperature responses. This shows that *ZF-ZHD* genes are stimulated by stress and that they are involved in stress-mediated pathways. Several studies have shown *ZF-HD/ZHD* gene participation in the case of abiotic/biotic stress. For example, the *ZF-HD/ZHD* gene family in *Arabidopsis*, tomato, cotton, grape, and Chinese cabbage was found to be involved in fighting various stress conditions, such as salt, drought, heat, and cold, by regulating stress-related hormones, such as ABA [1,27,42]. MYB and ARE help plants endure abiotic stress and are found in all the maize *ZHD* genes, especially *ZmZHD8* and *ZmZHD20*, in the case of MYB, and *ZmZHD9*, in the case of ARE (Figure 5). These findings suggest that these genes are responsible for maize tolerance to abiotic stress. *ZF-HD* is involved in a variety of biological activities in plants, including growth, development, and stress reduction [55]. Specific biological activities are determined by tissue-specific expression patterns [56]. The majority of *Arabidopsis ZF-HD/ZHD* genes are located in floral tissues, indicating that they play a role in floral development regulation [42]. *ZmZHD2*, *ZmZHD5*, and *ZmZHD21* exhibited greater expression patterns in floral tissues in this investigation (Figure 8), indicating that they may be involved in maize pollination [56,57]. The greater expression patterns of Cluster I and II genes in the heatmap for floral (unpollinated silk, endosperm after pollination, female spikelet, tip of ear primordium) and vegetative tissues (Figure 8) may have implications for their growth and development. *ZmZHD6* and *ZmZHD11* are the key genes in our study. They interact with other genes (Figure 7) and the *ZF-HD* protein dimerization region, including proteins, *ZF-HD* homeobox protein, and zinc finger-homeodomain protein (Appendix A).

MicroRNA was found to regulate the cellular responses of plants under stress conditions, such as salinity, cold, and dehydration [57,58,59,60]. Stress-responsive transcription factors (TFs) or functional genes are mainly targeted by several miRNAs [61]. Thus, miRNA may be involved in responses to stress conditions. Ten miR167 genes found in the maize *ZHD* genes (Appendix A) represent auxin response factor protein annotations found in maize [62]. This microRNA involved in maize shoot and leaf development enhances auxin response [62]. Another microRNA that regulates five *ZmZHD* genes (Appendix A) has not had its function elucidated yet. All of these results provide a valuable foundation for further future molecular investigations of *ZmZHD* genes and pave the way for the development of new varieties of maize.

## 4. Materials and Methods

### 4.1. Gene Retrieval and Sequence Analysis of the ZHD Gene Family in Maize

A BLASTp database search for *Z. mays* ZF dimers (PF04770) (sequence: *vrYreClrNhaaslGghavDGCgeFmasgeegtaeaLkCaaCgCHrnFHrree*) against *Arabidopsis*, sorghum, and rice was run to identify *ZF-HD* genes. For greater accuracy, ZF dimer domains (PF04770) were extracted from the Pfam database (http://pfam.xfam.org) accessed on 1 September 2022, and used to ensure the presence of this domain in the selected *ZF-HD/ZHD* genes, with an E-value < 1 × 10^−5^. Genome sequences, coding sequences (CDSs), and proteins sequences of *Z. mays* were retrieved from the Maize Genome Database (MaizeGDB; https://www.maizegdb.org/) accessed on 1 September 2022. Twenty-one *ZmZHD* gene domains were predicted using the Pfam database [63] and the SMART conserved domain search tool [64,65]. To estimate physiochemical properties, such as molecular weight (MW), isoelectric point (PI), and grand average of hydropathicity (GRAVY), the ExPASyProtParam tool (http://www.expasy.org/protparam/) accessed on 1 September 2022, subcellular localization of the retrieved genes, and Cell-Ploc 2.0 were used (Chou and Shen, 2010), respectively. *Arabidopsis* (15), the rice indica group (14), foxtail millet (16), sorghum (14), barley (11), purple false brome (*Brachypodium distachyon*) (21), and Heller’s rosette grass (*Dichanthelium oligosanthes*) *ZF-HD* protein sequences (Appendix A) were retrieved from different databases, such as the TIAR database (https://www.Arabidopsis.org/) accessed on 1 September 2022, the iTAK database [66], and the Plant TFDB database [67]. The HMMER web server and the InterPro online tool (http://www.ebi.ac.uk/interpro/) accessed on 1 September 2022 were used for the confirmation of the ZF domains of these protein sequences [63].

### 4.2. Sequence Alignment and Phylogenetic Tree Construction

The amino acid sequences extracted from maize (21), *Arabidopsis* (15), the rice indica group (14), foxtail millet (16), sorghum (14), barley (11), purple false brome (*B. distachyon*) (21), and Heller’s rosette grass (*D. oligosanthes*) were aligned using the ClustalX v2.1 multiple sequence alignment tool [68] and then exported to Genedoc for further elaboration of these protein sequences (https://www.nrbcs.org/gfx/genedoc/ebinet.htm) accessed on 1 September 2022 [69]. In the MEGA11.0 program, a phylogenetic tree was built using the neighbor-joining (NJ) technique with 1000 bootstrap replicates, and the tree was updated using iTOL (https://itol.embl.de/) accessed on 1 September 2022 [65,70].

For the following reasons, we favored the neighbor-joining method above others. The basic objective was to reconstruct phylogenetic trees using evolutionary distance information. The idea behind this approach is to identify the out pairs at each level of clustering, starting with a star-shaped tree, that have the shortest overall branch length. Using this technique, the branch lengths and topology of a parsimonious tree can be easily determined. This method’s primary goal is to establish relationships between sequences based on their genetic distances; however, the evolutionary model does not account for this. We checked and rechecked the ZF dimer (PF04770) domain in both the HMMER and Conserved Domains Database (CDD) and the Resources-NCBI database for confirmation. Both are good materials for bioinformatics, with advantages and disadvantages. We used BLASTp and HMMER for different purposes. To discover our chosen gene family sequences and to check the published data, we performed a BLASTp search of our domain sequences against the genome databases or annotation projects for the chosen plant species.

### 4.3. Chromosomal Locations, Motif Compositions, and Exon and Intron Distributions

The chromosomal locations of the 21 genes identified on the 10 maize chromosomes were mapped using the MapGene2Chrom web v2 web tool [71], and information was gleaned from Maize GDB (https://www.maizegdb.org/) accessed on 1 September 2022. The positions of the conserved motifs of these 21 ZHD protein sequences were analyzed by setting a maximum width of 50, a minimum width ≥6, and a motif number of 10; other parameters were set to default in the online MEME tool (http://meme-suite.org/) accessed on 1 September 2022 [72]. For exon and intron distributions, the gene structure display server (GSDS) web tool [43] was used to align genomic sequences with the CDSs of the 21 maize *ZHD* genes.

### 4.4. Cis-Acting Elements and Functional Prediction

Putative cis-elements in maize *ZHD* genes of about 5 to 10 bp were retrieved using the Plant CARE (http://bioinformatics.psb.ugent.be/webtools/plantcare/html/) accessed on 1 September 2022 [73] web-based tool. The 2000 bp upstream sequence for each gene from the start codon was extracted from Phytozome 13 (https://phytozome-next.jgi.doe.gov/) accessed on 1 September 2022 for cis-regulatory element extraction, as the upstream region contained cis-elements that bound the transcription factors that regulate target genes [74].

### 4.5. Synteny Analysis of ZHD Proteins and ks/ka Ratios

Protein sequences of *Z. mays* were compared with protein sequences from the rice indica group, sorghum, and foxtail millet using TBtools software. Synteny relationships and duplication events among the *ZHD* proteins were predicted using MCScan [75], and the findings about gene duplications with gene pairs were used to identify duplications in the *ZmZHD* genes of maize along with those of several monocots and dicots, the results being visualized in TBtools [76]. An NCBI BLAST search was run considering 80% sequence similarity against each of the maize ZHD proteins to determine gene duplications [77]. The protein sequences of duplicated gene pairs were first aligned in Clustal Omega [78]. Then, the sequence alignments of proteins and their associated cDNA sequences were used, by means of the PAL2NAL online tool, to determine the relevant codon alignments [79]. Finally, Ks and Ks values were estimated using PAML’s CODEML software and the generated codon alignments [36]. The synonymous substitution rate (Ks), nonsynonymous substitution rate (Ka), and Ka/Ks ratio were calculated for homologous gene pairs using Ka/Ks Calculator2.0 [46]. The equation T = Ks/2λ (where λ = 6.5 × 10^−9^) was used to compute evolutionary divergence periods within the *ZHD* gene family.

### 4.6. Protein–Protein Interaction and Z. mays L. RNA-Sequencing Data Analysis

Protein–protein interactions were assessed with the aid of the STRING database, with a medium (0.400) confidence level, to determine the interrelationships among the 21 *ZmZHD* proteins. The Expression Atlas database (https://www.ebi.ac.uk/gxa/home) accessed on 1 September 2022 was used to obtain *ZmZHD* RNA-sequencing data [80], which were previously collected and analyzed by Walley et al. [81]. We collected data for a total of 16 tissues of *Z. mays* L.: unpollinated silk (US), vegetative meristem (VM), pericarp and aleurone (PA), embryo after pollination (EAM), endosperm after pollination (EAP), internode (IN), mature leaf (ML), female spikelet (FL), primary root (PR), secondary root (SR), root differentiation zone (RDZ), root elongation zone (REZ), stomatal divisional zone of the leaf (SDZF), tip of ear primordium (TED), germinated embryo (GE), and growth zone of the leaf (GZL) tissues. In the acquisition of expression data, the fragments per kilobase of exon model per million mapped reads (FPKM) unit was utilized. The FPKM values were transferred to log2 (FPKM + 1) form and then a heatmap was built using the pheatmap package in Rstudio [82].

### 4.7. MiRNA Target Site Prediction

To determine the target sites of the 21 genes in the *ZmZHD* gene family in maize, first, mature miRNA was retrieved from the PmiREM server (https://www.pmiren.com/) accessed on 1 September 2022. Then, the CDSs of the 21 genes were searched against mature miRNAs using the online server tool PsRNA (https://www.zhaolab.org/psRNATarget/) accessed on 1 September 2022, with the default parameters [83]. The linkages between the predicted miRNAs were constructed using Cytoscape software (https://www.omicshare.com/tools/) accessed on 1 September 2022 [84].

## 5. Conclusions

In this study, we divided the 21 *ZHD* genes in maize (*Z. mays*) into two groups through phylogenetic analysis. Based on evolutionary research, these proteins were classified into seven subgroups. Exon–intron arrangements, motifs, and cis-acting regions were all comparable for the genes grouped. These *Z. mays ZHD* genes may play a role in biological processes and environmental stress control, according to the promoter and expression profiles of tissue-specific RNA sequencing studies. This study provides a foundation for exploring the roles of these genes in stressful environments and investigating their molecular mechanisms.

## Figures and Tables

**Figure 1 genes-13-02112-f001:**
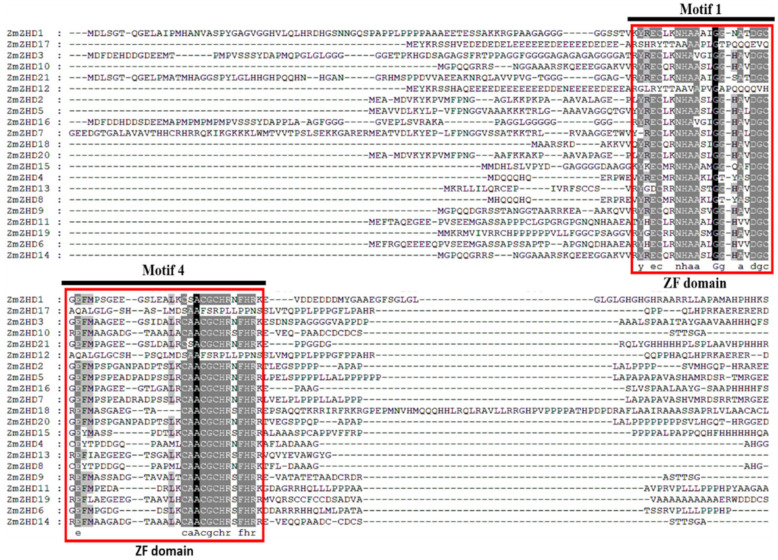
Multiple sequence alignment of the conserved domains of the members of the *ZmZHD* gene family in maize. Motifs 1 and 4 represent ZF domains.

**Figure 2 genes-13-02112-f002:**
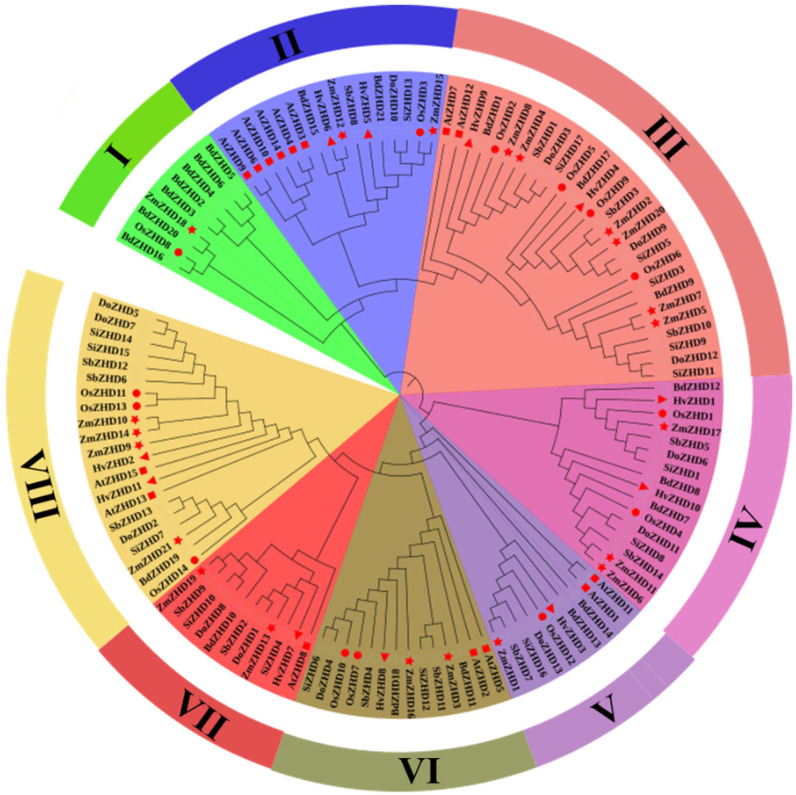
Phylogenetic analysis of full-length *ZHD* protein sequences from *Z. mays* (Zm, maize), *Arabidopsis thaliana* (At, *Arabidopsis*), *Oryzasativa* (Os, rice), *Sorghum bicholor* (Sb, sorghum), *Brachypodiumdistachyon* (Bd, purple false brome), and *Dichantheliumoligosanthes* (Do, Heller’s rosette grass). Red circles, red checkmarks, and red triangle, red rectangular, and red stars indicate *Arabidopsis*, and rice sequences, maize respectively.

**Figure 3 genes-13-02112-f003:**
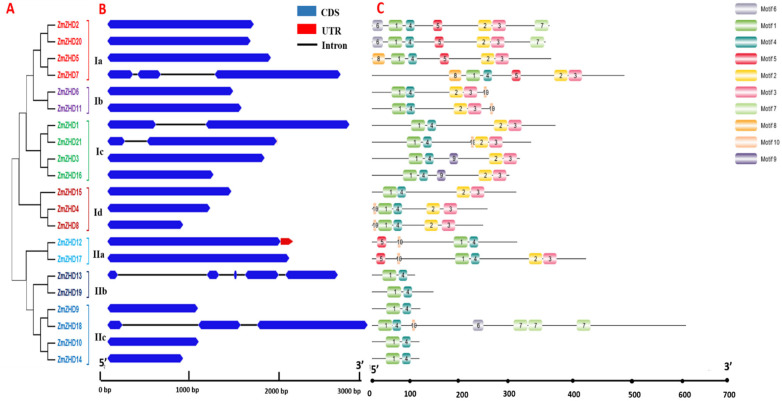
The phylogenetic relationships, conserved motifs, and gene structures of *ZmZHD* proteins and *ZmZHD* genes. (**A**) A maximum likelihood (ML) phylogenetic tree of the maize proteins was constructed from full-length sequences in MEGA 11.0 with 1000 bootstrap replicates. (**B**) The gene structures of the *ZmZHD* genes include introns (black lines), exons (blue rectangles), and untranslated regions (UTRs, red rectangles). The scale bar indicates 0.5 kb. (**C**) Distribution of conserved motifs in the *ZmZHD* proteins. The colored boxes represent Motifs 1–10. The scale bar indicates 100 aa.

**Figure 4 genes-13-02112-f004:**
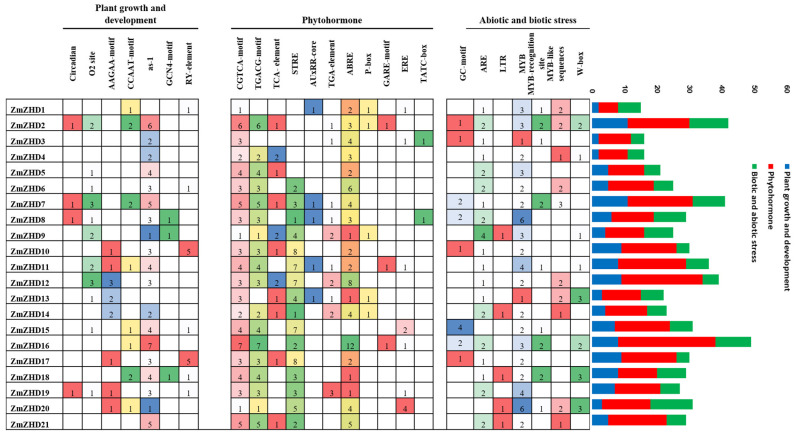
Cis-acting elements identified in the promoter regions of *ZmZHD* genes.

**Figure 5 genes-13-02112-f005:**
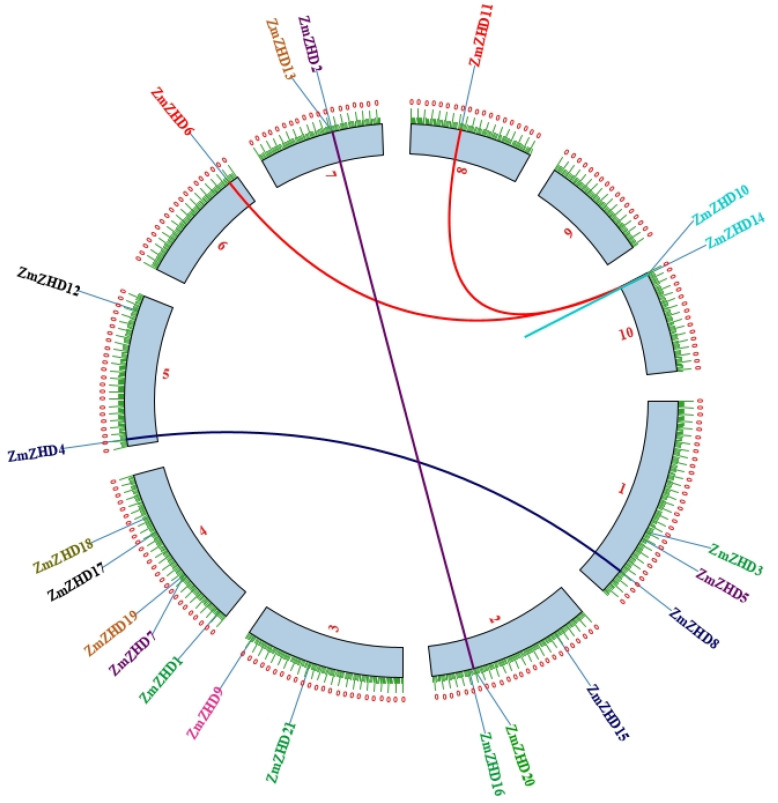
Chromosomal distribution and inter-chromosomal relationships of *ZmZHD* genes. Red lines connect duplication gene pairs between *ZmZHD10* and *ZmZHD11* and between *ZmZHD6* and *ZmZHD10*; blue lines connect duplication gene pairs between *ZmZHD4* and *ZmZHD8*, and violet lines connect duplication gene pairs between *ZmZHD2* and *ZmZHD20*.

**Figure 6 genes-13-02112-f006:**
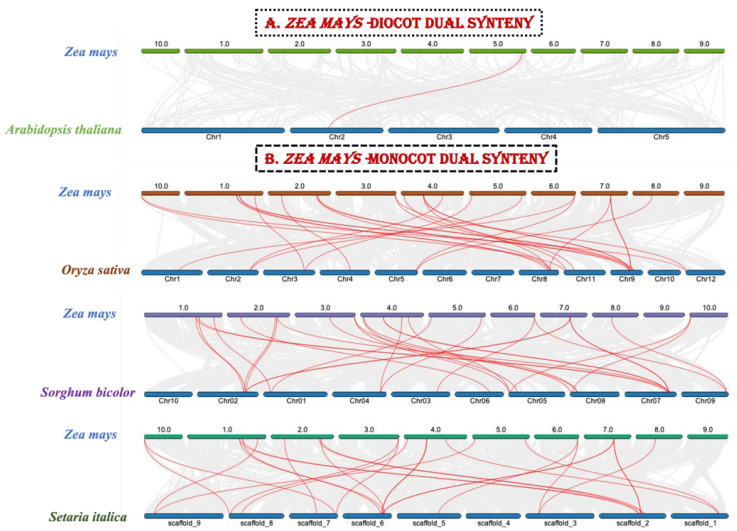
Synteny analysis of the maize genome with one monocot (**A**) and three dicot (**B**) plant genomes. The gray lines represent aligned blocks between the paired genomes, and the red lines indicate syntenic *ZHD* gene pairs. We performed both dual synteny and specific gene family synteny analyses for the maize genome, this being one of the most important fields in comparative genomic analysis, as it is the basis of evolutionary studies at both the gene and genome levels. We used the species-specific gene family protein sequences, but not in the synteny analysis, as most of the causes have not been properly studied, for instance, the chromosomal gene positions are quite enigmatic. Insofar as we could find well-researched sequences online, we retrieved and dual-synteny-analyzed them. In addition, a specific gene family from the maize genome synteny analysis was studied to comprehend duplication occurrences and internal evolutionary processes.

**Figure 7 genes-13-02112-f007:**
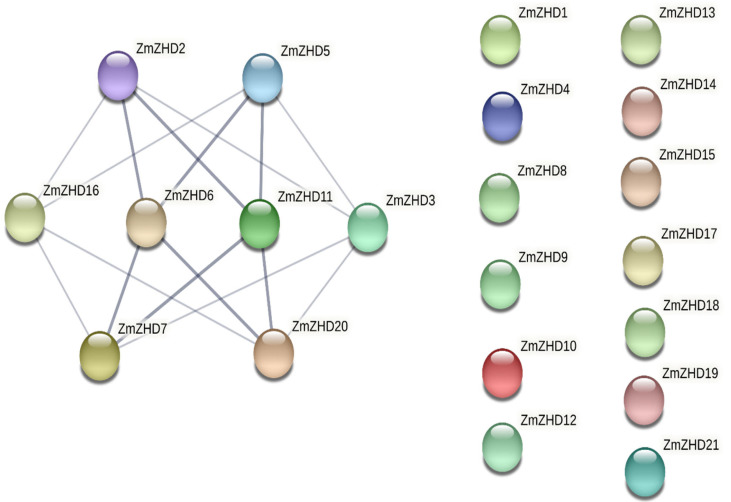
Interaction network of the *ZHD* proteins in *Z. mays* L. Deep ash-colored lines indicate the highest level of confidence (0.900), and faded ash-colored lines indicate a medium level of confidence (0.400). Lineless proteins do not have any relationship with other proteins. Protein–protein interactions (PPIs) play a crucial role in cellular functions and biological processes, including cell–cell interactions and metabolic and developmental control in all organisms. We performed in silico protein–protein interaction analyses within the family for phylogenetic profiling and to identify structural patterns and homologous pairs, intracellular localizations, and post-translational modifications among the proteins. Furthermore, we considered the interaction and involvement of major signalling or stress pathways, though we avoid discussion of these subjects due to their complexity.

**Figure 8 genes-13-02112-f008:**
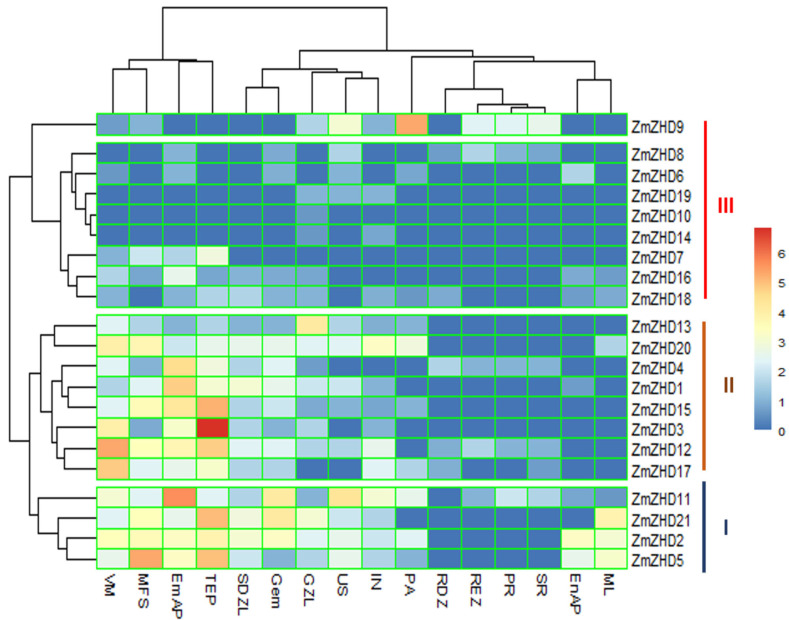
Heatmap showing the expression levels of *ZmZHD* genes in different tissues. Normalized Log2 (FPKM + 1) values are plotted against respective tissues. Tissue name abbreviations are as follows: unpollinated silk (US), vegetative meristem (VM), pericarp and aleurone (PA), embryo after pollination (EmAP), endosperm after pollination (EnAP), internode (IN), mature leaf (ML), mature female spikelet (MFS), primary root (PR), secondary root (SR), root differentiation zone (RDZ), root elongation zone (REZ), stomatal divisional zone of the leaf (SDZL), tip of the primordium (TEP), germinated embryo (Gem), growth zone of the leaf (GZL).

**Figure 9 genes-13-02112-f009:**
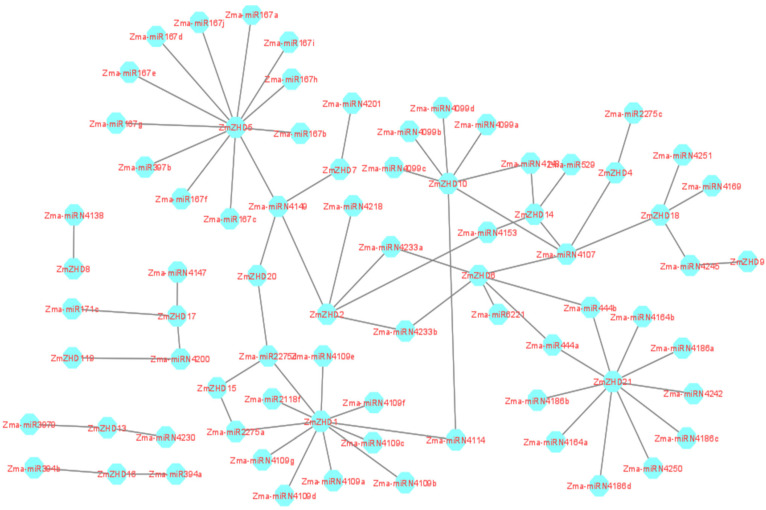
Picture of the regulatory network relationships between the putative miRNAs and their targeted maize *ZHD* genes.

**Table 1 genes-13-02112-t001:** Detailed information about the *ZmZHD* genes and corresponding proteins in *Z. mays* L.

Gene Name	Gene ID	Chromosome Location				Protein Length (aa)	Mol. Wt. ^1^KDA	pI ^1^	GRAVY ^1^	Exon	Intron	Subcellular Location
		Number	Start	End	Length							
*ZmZHD1*	GRMZM2G068330	4	11278503	112881332	250330460	382	39,774.87	8.13	−0.58	2	1	Nucleus
*ZmZHD2*	GRMZM2G161315	7	112902252	1129906963	185808916	370	38,611.45	8.21	−0.524	1		**Cell wall, nucleus**
*ZmZHD3*	GRMZM2G346920	1	200895999	200900836	308452471	361	37,825.61	7.03	−0.515	1		**Chloroplast, nucleus**
*ZmZHD4*	GRMZM2G425236	5	10221783	10225985	226353449	240	24,975.74	7.16	−0.629	1		Nucleus
*ZmZHD5*	GRMZM2G438438	1	212576317	212581228	308452471	373	38,662.73	7.21	−0.31	1		Nucleus
*ZmZHD6*	GRMZM2G414844	6	166374440	166378910	181357234	242	26,692.75	8.8	−0.985	1		Nucleus
*ZmZHD7*	GRMZM2G353734	4	85319366	85325092	250330460	526	55,497.21	9.02	−0.575	3	2	Nucleus
*ZmZHD8*	GRMZM2G423423	1	269145949	269149835	308452471	231	24,075.72	8.35	−0.687	1		Nucleus
*ZmZHD9*	GRMZM2G353076	3	230480803	230481862	238017767	100	10,401.41	8.93	−0.608	1		Nucleus
*ZmZHD10*	GRMZM2G470974	10	2632775	2633842	152435371	98	10,100.06	**6.87**	−0.628	1		Nucleus
*ZmZHD11*	GRMZM2G328438	8	73654879	73656447	182411202	254	27,689.12	8.51	−0.805	1		Nucleus
*ZmZHD12*	GRMZM2G417229	5	201282254	201284278	226353449	302	32,353.12	**6.95**	−0.746	1		Nucleus
*ZmZHD13*	GRMZM2G071112	7	112658777	112661470	185808916	**89**	9802.04	8.16	−0.361	5	4	Nucleus
*ZmZHD14*	GRMZM2G172586	10	2639262	2640147	152435371	98	10,113.1	7.59	−0.628	1		Nucleus
*ZmZHD15*	GRMZM2G089619	2	50140925	50142374	243,675,191	300	31,130.79	**6.72**	−0.437	1		Nucleus
*ZmZHD16*	GRMZM2G389379	2	188271896	188273136	243,675,191	286	29,912.64	**6.96**	−0.485	1		Nucleus
*ZmZHD17*	GRMZM2G069365	4	160153804	160155930	250330460	446	48,227.78	**6.64**	−0.793	1		Nucleus
*ZmZHD18*	GRMZM2G462417	4	185816491	185819532	250330460	**655**	**71,591.15**	9.01	−0.046	3	2	Nucleus
*ZmZHD19*	GRMZM2G370863	4	85758755	85759432	250330460	127	13,351.19	7.53	−**0.138**	1		Nucleus
*ZmZHD20*	GRMZM2G051955	2	181822882	181824556	243675191	361	37,714.29	7.73	−0.578	1		Cell wall, nucleus
*ZmZHD21*	GRMZM5G821755	3	136934215	136936196	238017767	331	35,115.48	8.57	−0.593	2	1	Nucleus

^1^ Mol. Wt.—Molecular weight, pI—Iso-electric point, GRAVY—Grand average of hydropathy.

## Data Availability

Not applicable.

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
