# Peer review of "Genome-Wide Identification and In Silico Analysis of ZF-HD Transcription Factor Genes in Zea mays L."

_genes, 2022, doi:10.3390/genes13112112_

Round 1

Reviewer 1 Report

The paper is a very extensive and detailed analysis, and the work was carefully designed and performed. Overall, I am positive about this manuscript, although I feel that some points have to be remedied - most can be addressed by careful text work. Please see the specific comments below:

Abstract:

L31: Please avoid to start the manuscript with abbreviations or writing full name.

L31: Please add: that are

Introduction

L55-56: Please recheck the sentence.

L59: Either write HD or write homeodomain.

L69-73: Please restructure the sentence to make it more clear.

L76-79: Please revise the sentence.

L100: devloped? developed countries?

L102: Please revise the sentence.

Results:

L123: Please revise the punctuation as follows:

"maize," rather than maize number,

L124: add "Whereas" instead of "where"

L128-129: The sentence is unclear "while the rest of them showed higher than 7 (Table1), which 128 means those which have less than 7 indicating acidic. "

Fig.1 Please improve the Figure and use same font throughout the figure.

Fig.6 Why the synteny analysis does not include the other genomes that are already used in phylogenetic analysis? Please elaborate

Fig.7 Why did you check the in-silico protein-protein interaction within the family? Why you did not check its interaction and involvement in any major signaling or stress pathway?

Discussion:

L291: 296 Please, restructure the sentence carefully by considering the punctuation.

L302: The sentence is unclear.

L306: have similar structure for what? in all plants or all cereals? Please clarify the sentence.

L313-316: Please elaborate the statement with some more literature.

L323: Please elaborate the statement "similar motifs present in same subclass".

L325-330: Please re-write this sentence with more clear rationale and add more relevant references.

L336-337: The sentence in unclear.

Material and Methods:

Elaborate the reasons for choosing the False brome and rosette grass?

L371: Why did you prefer BLASTp over HMMER for sequence search?

L398: Please elaborate why did you Neighbour-joining technique over other method?

Reference

Recheck the whole bibliography for formatting mistakes and make it uniform according to journal foramat.

Author Response

SN

Reviewers’ Comments

Response

1

Please avoid to start the manuscript with abbreviations or writing full name.

Abbreviations were replaced

2

Please add: that are

Instruction had been following

3

Please recheck the sentence.

The sentence was rephrased

4

L59: Either write HD or write homeodomain.

HD was replaced with HomeDomain

5

L69-73: Please restructure the sentence to make it more clear.

The sentence was rephrased

5

L100: devloped? developed countries?

The sentence was rephrased, Zea mays L. is the most extensively grown cereal crop in Africa and South America. Nowadays, it is getting popular in developing countries like Bangladesh, and other developed countries

6

L102: Please revise the sentence.

The instruction was followed  

7

L123: Please revise the punctuation as follows:

"maize," rather than maize number,

Instruction has been followed

8

L124: add "Whereas" instead of "where"

Instruction has been followed

9

L128-129: The sentence is unclear "while the rest of them showed higher than 7 (Table1), which 128 means those which have less than 7 indicating acidic. "

Gene ZmZHD10, ZmZHD12, ZmZHD15, ZmZHD16, and ZmZHD17 are associated with less than 7 theoretical pI values, while the rest of them showed higher than 7 (Table1), which shows the values where the amino acids can be neutral

1o

Fig.1 Please improve the Figure and use same font throughout the figure.

This suggestion has been followed.

11

‘Fig.6 Why the synteny analysis does not include the other genomes that are already used in phylogenetic analysis? Please elaborate

We performed both dual synteny and a specific gene family of maize genome synteny analysis which it is one of the most important fields in comparative genome analysis as is the basis of evolutionary studies at both the gene and ge-nome levels. We used the species-specific gene family protein sequences but not in the synteny analysis as most of the causes are not properly studied for instance their chromosomal gene position is quite enigmatic. As far as we could find well-researched sequences online, we retrieved and dual synteny-analyzed them. In addition, a specific gene family of the maize genome synteny analysis was studied to comprehend duplication occurrences and internal evolutionary processes.

12

Fig.7 Why did you check the in-silico protein-protein interaction within the family? Why you did not check its interaction and involvement in any major signaling or stress pathway?

Protein-protein interactions (PPIs) play a crucial role in cellular functions and biological processes, including cell-to-cell interactions and metabolic and developmental control in all organisms. We did in-silico protein-protein interaction within the family to find out phylogenetic profiling, identifying structural patterns and homologous pairs, intracellular localization, and post-translational modifications, among themselves. Furthermore, we thought about to see interaction and involvement in any major signaling or stress pathway as you asked. We avoided this part for some complexity.

13

L291: 296 Please, restructure the sentence carefully by considering the punctuation

Instruction has been followed

14

L302: The sentence is unclear.

L306: have similar structure for what? in all plants or all cereals? Please clarify the sentence.

Lines have been rephrased

15

L313-316: Please elaborate the statement with some more literature.

Sentence has been changed

16

L325-330: Please re-write this sentence with more clear rationale and add more relevant references.

Suggestion has been followed

17

L336-337: The sentence in unclear.

Lines have been rephrased

18

Elaborate the reasons for choosing the False brome and rosette grass?

To compare monocots with monocots

19

L371: Why did you prefer BLASTp over HMMER for sequence search?

We checked and rechecked the ZF dimer (PF04770) domain in both the HMMER and Conserved Domains Database (CDD) and the Resources - NCBI database for confirmation. Both are good materials for bioinformatics with advantages and disadvantages. We used BLASTp and HMMER in different purposes. To discover our chosen gene family sequenc-es and to check the published data, we performed a BLASTp search of our domain se-quence against the genome databases or annotation projects of the chosen plant species.

20

L398: Please elaborate why did you Neighbour-joining technique over other method?

For the following reasons, we favored the Neighbor-joining method above others. The basic objective is to reconstruct phylogenetic trees using evolutionary distance infor-mation. The idea behind this approach is to identify OTU pairs at each level of clustering, starting with a star-shaped tree, that have the shortest overall branch length. Using this technique, the branch lengths and topology of a parsimonious tree can be easily deter-mined. This method's primary goal is to establish relationships between sequences based on their genetic distance, however the evolutionary model does not account for this.

Reviewer 2 Report

The manuscript is about the detailed In-Silico analysis of the Zing Finger Transcription factor and bears the potential to get published in the Genes. However, several points in the manuscript need to be reconsidered. Therefore, I suggest a major revision of the manuscript.

Title:

The Title should be changed to “Genome-Wide Identification and In-Silico analysis of ZF-HD Transcription Factor Genes in Zea mays. L”

Introduction:

Please restructure the introduction, especially the first paragraph, to build a more interesting background and rationale for your study. I suggest starting from the ZFs, rather than stresses because the study is in-silico.

Results:

Results are well-organized as characteristic of in-silico studies. However, the write-up of results needs extensive revision as at some points, due to poor punctuation or sentence structure, the meaning of sentences remains unclear.

Some parts, e.g., include all the mentioned species, while others include only a few species with maize. Please elaborate.

For example, L128-129. Check all the results sections for unclear sentences and grammatical errors.

Methods:

There are many different platforms for the same analysis. Therefore, I suggest the authors add the reason for using a particular platform, where possible.

 In 4.1, also add the reason to choose the particular species for comparison.

Discussion:

In the first paragraph, please add some more literature about the roles and relate them to the present study.

L302-308 Please revise the sentence to give clear meaning and reconsider the references cited in the discussion. Please avoid repetition of results, rather you should explain and justify your results.

L311-313 Please elaborate on these points in more detail to clarify the concept about lack of intron and relate it with evolution.

In paragraph number 4, please add some more detail about the role of cis-elements and relate them with stress and your studies.

In the discussion section, add a separate paragraph that should discuss the expression of ZmZHD in different tissue and its significance.

Moreover, I found many grammar errors throughout the manuscript, but I did not point them out in detail because that is the authors' responsibility. Please correct them.

Author Response

1

’ The Title should be changed to “Genome-Wide Identification and In-Silico analysis of ZF-HD Transcription Factor Genes in Zea mays. L”

the title was changed

2

Please restructure the introduction, especially the first paragraph, to build a more interesting background and rationale for your study. I suggest starting from the ZFs, rather than stresses because the study is in-silico.

Changes were ,made according to suggestion

3

Results are well-organized as characteristic of in-silico studies. However, the write-up of results needs extensive revision as at some points, due to poor punctuation or sentence structure, the meaning of sentences remains unclear.

Some parts, e.g., include all the mentioned species, while others include only a few species with maize. Please elaborate.

For example, L128-129. Check all the results sections for unclear sentences and grammatical errors.

 This sentences were reviewed and revisions were made

4

In the first paragraph, please add some more literature about the roles and relate them to the present study.

 Suggestion has been followed

5

In the first paragraph, please add some more literature about the roles and relate them to the present study.

It has been written.

6

L311-313 Please elaborate on these points in more detail to clarify the concept about lack of intron and relate it with evolution.

It has been written.

7

In paragraph number 4, please add some more detail about the role of cis-elements and relate them with stress and your studies.

It has been written.

Round 2

Reviewer 2 Report

No more comment